**Data Availability Statement:** All relevant data are within the paper and its Supporting Information files.

# A retrospective evaluation of the effect of oclacitinib (Apoquel) administration on development of surgical site infection following clean orthopedic stifle surgery

**Alyssa K. Motz** [ID]*◉, **Lindsay L. St. Germaine**◉, **Daniel E. Hoffmann**‡, **Jed Sung**‡

Surgery Department, Veterinary Specialists & Emergency Services, Rochester, NY, United States of America

◉ These authors contributed equally to this work.
‡ DEH and JS also contributed equally to this work.
* alyssakmotz@gmail.com

## Abstract

The aim of this study was to determine the effect of oclacitinib (Apoquel) on development of surgical site infections in canines following clean orthopedic stifle surgery. Medical records of dogs undergoing unilateral, clean orthopedic stifle procedures were retrospectively examined for development of post-operative surgical site infections. Data collected for statistical analysis included age, sex, body weight, current medications, anesthesia and surgery times, white blood cell count, and neutrophil count. Surgical site infections were identified in 8.7% (34/390) of stifle procedures– 8.0% (29/364) in dogs not treated with oclacitinib and 19.2% (5/26) in dogs treated with oclacitinib (p = 0.053). There was a significant difference in development of surgical site infection in dogs with longer anesthesia times (p = 0.003) and higher body weights (p = 0.037). Dogs being treated with oclacitinib at the time of clean, orthopedic stifle surgery did not have a significantly higher incidence of surgical site infections. However, client education regarding risk of infection and increased patient monitoring post-operatively are recommended, especially in patients with increased body weight or longer anesthetic times.

## Introduction

Stifle surgeries to address cranial cruciate ligament rupture or patellar luxation are the most common orthopedic procedures performed on canine patients. Surgical site infection (SSI) following these procedures is a known complication. Previous studies have found SSI rates of 2.3%-25.9%, 2.6%-15.4%, 4.2–17.3%, and 2.2%-10.3% after tibial plateau leveling osteotomy (TPLO), tibial tuberosity advancement (TTA), lateral fabellar suture (LFS), and medial patellar luxation (MPL) correction surgeries, respectively [1–20]. Increased risks of surgical site infections have been found to be multifactorial, with increased risk found to be associated with age, weight, duration of anesthesia and surgery, among others [1, 5, 8, 10–12, 21–23]. Minimizing surgical site infection rates is important for optimal patient healing, improved client

**Funding:** The authors received no specific funding for this work.

**Competing interests:** The authors have declared that no competing interests exist.

satisfaction, and reduction of additional client costs [24]. Importantly, use of antibiotics is considered a driving force behind the emergence of strains of multi-drug resistant bacteria and efforts should be made to minimize the need of additional antibiotic administration.

Atopic dermatitis treated with immunomodulating therapies has become prevalent in our profession, including a large subset of patients in need of stifle surgery. Oclacitinib (Apoquel) is a selective JAK1 inhibitor used to treat atopic dermatitis by immune system modulation through blocking of proinflammatory cytokine signaling [25]. Oclacitinib has been noted to cause transient neutropenia and may increase susceptibility to infection [25–27]. To the authors' knowledge, no studies have evaluated the incidence of SSI in dogs concurrently receiving oclacitinib for treatment of atopic dermatitis at the time of a clean orthopedic surgery.

The objective of this paper is to evaluate the rate of surgical site infection in dogs on oclacitinib therapy undergoing clean stifle orthopedic procedures. The authors hypothesize there will be no significant difference in the rate of surgical site infections after clean orthopedic surgery between dogs receiving concurrent oclacitinib therapy versus dogs who were not.

## Material and methods

### Data collection

The medical records of dogs undergoing unilateral, clean, orthopedic stifle procedures between January 2018 and February 2021 were reviewed. Data collected for each patient included age at time of surgery, sex, body weight, current medications, comorbidities, history of skin infections, previous orthopedic procedures, complete blood count, chemistry profile, procedure performed (TPLO, TTA, LFS, MPL correction), additional surgical procedures, incision closure with staples or suture, surgeon, total anesthesia time, surgery time, induction agent, intraoperative antibiotics, post operative medications, incisional evaluations at 2 and 8 week rechecks, and development of surgical site infections. Dogs were included if there was a complete medical record and were reviewed to 90 days following surgery. Dogs were excluded if there was an incomplete medical record, if bilateral single session procedures were performed, if the procedure was a revision surgery or any other additional procedures were done during the same anesthetic event. Dogs were also excluded if they had concurrent skin infection anywhere on their body, received a course of prophylactic post-operative antibiotics for any reason, had a previously diagnosed endocrinopathy, or any other previously identified comorbidities. Second side procedures were not included for analysis. For dogs receiving oclacitinib, medical records were evaluated to confirm current oclacitinib dosing at the time of surgery. Oclacitinib was prescribed by patients' referring veterinarians according to the labeled dosing guidelines (0.4–0.6mg/kg). All patients were on once daily maintenance dosing.

Approval of an animal research ethics committee or patient/owner consent was not obtained due to the retrospective nature of this study. No animals received altered treatments or procedures for the purpose of this study Data was collected from stored medical records retrospectively.

Surgical site infection events were identified from the medical record using guidelines adapted from Centers for Disease Control and Prevention (CDC) [28]. Surgical site infections were categorized as superficial, deep, or organ/space. A superficial SSI was defined as a surgical site abnormality involving the skin or subcutaneous tissues within 30 days of the procedure and the presence of one or more of the following: purulent discharge, organism identified on bacterial culture, incisional dehiscence, one sign of infection (pain or tenderness, localized swelling, redness or heat), or diagnosis of surgical site infection by a veterinarian [28]. A deep SSI was defined as a surgical site abnormality involving the deep soft tissues of the incision

(fascial or muscle) within 90 days and one or more of the following: purulent discharge, deep incisional dehiscence, organism identified on culture of deep soft tissues, pyrexia, or signs of localized pain or tenderness [28]. An organ/space SSI was defined as surgical site abnormality involving the organ or space (stifle joint) opened during surgery within 90 days and one or more of the following: purulent discharge, organism identified on culture, or evidence of infection on gross anatomic or histologic examination [28]. Dogs were included if signs of surgical site infections were identified within 90 days of surgery.

## Procedure

Dogs were placed under general anesthesia using a protocol custom designed by the surgeon. The surgical leg was clipped and hung in standard fashion and the skin was defatted using 70% isopropyl alcohol then scrubbed with 5% chlorohexidine for 5 minutes prior to transport to the operating room. A final sterile preparation of the limb was performed in the operating room with the limb hung, again using a 5% chlorohexidine scrub for 5 minutes followed by 70% isopropyl alcohol. Perioperative anesthetic records confirmed all dogs received cefazolin at 22mg/kg IV within 60 minutes of the first incision and then every 90 minutes thereafter until the end of anesthesia. The required procedure (TPLO, TTA, LFS, or MPL correction) was performed by a board-certified surgeon or surgical resident under direct supervision by a board-certified surgeon. Following the procedure, triple antibiotic ointment was applied once over the incision, followed by a telfa pad secured with an adhesive bandage as part of standard hospital protocol prior to transport for post-operative radiographs. A full limb modified Robert Jones bandage was placed for 24 hours once appropriate implant position was confirmed on radiographs. All patients had custom designed post-operative analgesia plans which included a combination of opioid and non-steroidal anti-inflammatory medications. Oclacitinib was not administered to patients on the day of surgery. Pet owners were instructed to resume administration of oclacitinib after discharge the day following surgery.

All patients were evaluated at two weeks and eight weeks following surgery, unless owner concerns required additional evaluations. All incisions were evaluated and assessed for signs of surgical site infections (pain or tenderness, localized swelling, redness, heat, discharge, or dehiscence) at the two-week recheck. Additional rechecks were performed as needed until appropriate incisional healing was achieved. Two view orthogonal stifle radiographs were obtained at eight weeks following surgery to evaluate osteotomy healing (for MPL, TTA and TPLO). For LFS procedures, an eight week recheck examination was performed.

## Statistical analysis

The response variable was surgical site infection. This study evaluated surgical site infection and the following variables: treatment with oclacitinib and the continuous factors age, weight (kg), anesthesia time (minutes), surgery time (minutes), total white blood cell count, and neutrophil count. Analysis was by means of multivariate logistic regression. Multicollinearity was assessed by means of the variance inflation factor; VIF <2.5 were acceptable. Linearity was assessed by means of the Box Tidwell approach; the quadratic term was added to the model if nonlinear. All factors with univariate $P<0.30$ were initially added to the equation and deleted if $P>0.10$. Then all previously unused and deleted factors were added to the equation, one at a time, and not retained if $P>0.10$.

## Results

Of the 811 clean, orthopedic stifle procedures identified, 390 procedures on 390 dogs met the inclusion criteria. Of the 390 dogs included, 26 dogs (6.7%) were being treated with oclacitinib

at the time of surgery. Surgical site infections were identified in 34/390 (8.7%) dogs. The rate of surgical site infection in dogs not treated with oclacitinib was 8.0% (29/364), while the rate of surgical site infection in dogs treated with oclacitinib was 19.2% (5/26). This was not a significant difference (p = 0.064) in univariate analysis. In multivariate analysis, treatment with oclacitinib approached significance for development of an SSI (p = 0.053) and the odds of developing an SSI 2.944 times higher while on oclacitinib than if not. Aerobic cultures were obtained in 25/34 patients treated for surgical site infections, yielding 21/25 positive culture results.

There was no significant difference in age between dogs who did and dogs who did not develop surgical site infections (p = 0.470). There was also no significant difference in age between dogs who were and were not treated with oclacitinib at the time of surgery (p = 0.480).

There was a significant difference in weight (kg) between dogs who did and dogs who did not develop surgical site infections regardless of oclacitinib therapy (p = 0.037). In multivariate analysis, weight was close to being significantly associated with development of an SSI (p = 0.060). The odds of getting an SSI were found to be 2.9% higher for every 1 kg increase in body weight. However, there was no significant difference in weight between dogs who were and were not treated with oclacitinib at the time of surgery (p = 0.406).

There was a significant difference in anesthesia time between dogs who did (range: 157–280 minutes, median:195 minutes) and dogs who did not (range: 90–255 minutes, median:180 minutes) develop surgical site infections (p = 0.003). In multivariate analysis, anesthesia time remained significantly associated with development of an SSI (p = 0.004). For every 1 additional minute increase in anesthesia time, the odds of getting and SSI increased by 2.0%. However, there was no significant difference in anesthesia time between dogs who were and were not treated with oclacitinib at the time of surgery (p = 0.450).

There was a significant difference in surgery time between dogs who did (range:35–165 minutes, median:66 minutes) and did not (range:25–125 minutes, median:60 minutes) develop surgical site infections (p = 0.026). There was no significant difference in surgery time between dogs who were and were not treated with oclacitinib at the time of surgery (p = 0.794).

There was no significant difference in total white blood cell count between dogs who did (range:3.74–11.9 mm$^3$, median:7.8 mm$^3$) or did not (range:3.44–36.5 mm$^3$, median:7.4 mm$^3$) develop surgical site infections (p = 0.798). There was no significant difference in total white blood cell count between dogs who were and were not treated with oclacitinib at the time of surgery (p = 0.080).

There was no significant difference in neutrophil count between dogs who did (range:2.95–9.94 mm$^3$, median:5.3 mm$^3$) and did not (range:1.92–32.5 mm$^3$, median:5.2 mm$^3$) develop surgical site infections (p = 0.771). There was no significant difference between dogs who were and were not treated with oclacitinib at the time of surgery (p = 0.271).

Of the 34 SSIs identified, 26 (76.4%) were classified as superficial and 8 (23.6%) were classified as deep or organ/space. There was no significant difference in age (p = 0.694), weight (p = 0.069), surgery time (p = 0.075), total white blood cell count (p = 0.663), or neutrophil count (p = 0.474) between dogs who did not develop an SSI, developed a superficial SSI, or developed a deep or organ/space SSI. Dogs who developed superficial surgical site infections had significantly longer anesthesia times (range 157–280 minutes, median 195.5 minutes) than those who did not develop surgical site infections (range 90–255 minutes, median 180 minutes) (p = 0.005), however there was not a significant difference in anesthesia time between those who developed deep or organ/space infections and those who developed superficial infections (p = 0.706) or those who did not develop infection (p = 0.241). There was no significant effect in treatment with oclacitinib (p = 1.0) between dogs who did not develop an SSI, developed a superficial SSI, or developed a deep or organ/space SSI.

## Discussion

The purpose of this study was to identify if dogs undergoing clean, orthopedic stifle procedures while being treated with oclacitinib were at a higher risk for surgical site infections than those who were not treated with oclacitinib. The incidence of surgical site infections in this study population was 8.7%, which is within the previously reported ranges of SSIs following clean, orthopedic stifle procedures [1–20]. Statistically, there was no significant difference in incidence of surgical site infections between dogs treated with oclacitinib (19.2%) and those who were not (8.0%), however this approaches significance when using a multivariate analysis. The lack of statistical significance may be the result of type II error and may change with increased numbers of cases. Additionally, the odds ratio indicating dogs treated with oclacitinib have three times higher risk of developing an SSI should also be considered as support of this idea.

As an immunomodulating therapy, oclacitinib has been documented as having the potential to increase susceptibility to infection [25]. In previous studies, dogs on oclacitinib had decreased white blood cell count and neutrophil counts compared placebo groups and individual animals developed neutropenic leukopenia, but group means remained within the normal reference range [25–27]. This study had similar, although not significantly different, findings in that the median total white blood cell count for dogs on oclacitinib (7.1 x $10^9$/L) was lower than dogs who were not on oclacitinib (7.5 x $10^9$/L) and both median values were still within the normal reference range.

This study found increased risk of surgical site infection development associated with increasing body weight. The odds of getting an SSI were found to be 2.9% higher for every 1 kg increase in body weight. This odds ratio held true across comparison of all weight groups. This finding is in agreement with multiple previously published studies [5, 29–31].

Significant increased risk of surgical site infection was also found to be associated with increased anesthesia and surgery time. These risk factors have also been previously identified in numerous studies [1, 29–31]. In this study, for every 1 additional minute increase in anesthesia time, the odds of getting and SSI increase by 2.0%. Most orthopedic surgeries have additional anesthesia time devoted to pre- and/or post operative radiographs for surgical planning and evaluation of implant placement, respectively. Obtaining pre-operative planning radiographs prior to the day of surgery may help decrease the overall anesthesia time and risk of surgical site infection.

Cases were determined to be included as surgical site infections based on CDC guidelines. Updated guidelines have introduced procedure specific guidelines that indicate infections detected in human orthopedic surgeries beyond 90 days are not considered surgical site infections [28]. This criterion was adapted for the current study and thus only patients who developed signs of infection within 90 days of surgery were included. Additional cases of infection may have been identified with an expanded inclusion time period.

Limitations of the study are primarily related to its retrospective design. The potential for errors in data collection from medical records may lead to missed surgical site infections based on record descriptions or inclusion of patients with incisional inflammation as those with infections. Diagnosis of infection versus inflammation without a positive bacterial culture is subjective and may lead to more diagnosed SSIs as the perceived risk of treating inflammation as infection is less than that of missing a true infection by thinking it is inflammation. Conversely, it may also result in underreporting of SSI if a patient is seen by their primary veterinarian and that information is not relayed to the surgery service.

An additional limitation to this study is the difference in group population size. Dogs treated with apoquel comprised only 26 of 390 patients (6.7%) in this study. Comparing treatment groups of more equal sizes may have changed the statistical significance reported.

The findings of this study indicate that dogs being treated with oclacitinib at the time of surgery did not have a significantly higher incidence of surgical site infections following unilateral, clean, orthopedic stifle surgery. However, client education regarding risk of infection and increased patient monitoring post operatively are recommended, especially in patients with increased body weight or longer anesthetic times.

## Supporting information

**S1 Data.**
(XLSX)

## Author Contributions

**Conceptualization:** Alyssa K. Motz, Lindsay L. St. Germaine.

**Investigation:** Alyssa K. Motz.

**Project administration:** Lindsay L. St. Germaine, Daniel E. Hoffmann.

**Supervision:** Lindsay L. St. Germaine, Daniel E. Hoffmann, Jed Sung.

**Writing – original draft:** Alyssa K. Motz.

**Writing – review & editing:** Alyssa K. Motz, Lindsay L. St. Germaine, Daniel E. Hoffmann, Jed Sung.

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
