## [Decision Letter · Decision Letter 0]

28 Jun 2023

PONE-D-23-10920Effect of oclacitinib (Apoquel) administration on development of surgical site infection following clean orthopedic stifle surgeryPLOS ONE

Dear Dr. Motz,

Thank you for submitting your manuscript to PLOS ONE. After careful consideration, we feel that it has merit but does not fully meet PLOS ONE’s publication criteria as it currently stands. Therefore, we invite you to submit a revised version of the manuscript that addresses the points raised during the review process.

Please revise.

We look forward to receiving your revised manuscript.

Kind regards,

Academic Editor

PLOS ONE

Journal Requirements:

Reviewers' comments:

Reviewer's Responses to Questions

**Comments to the Author**

1. Is the manuscript technically sound, and do the data support the conclusions?

Reviewer #1: Yes

Reviewer #2: Yes

2. Has the statistical analysis been performed appropriately and rigorously? 

Reviewer #1: Yes

Reviewer #2: Yes

3. Have the authors made all data underlying the findings in their manuscript fully available?

Reviewer #1: Yes

Reviewer #2: No

4. Is the manuscript presented in an intelligible fashion and written in standard English?

Reviewer #1: Yes

Reviewer #2: Yes

5. Review Comments to the Author

Reviewer #1: Excellent work. Just a few additions to be considered:

1. Please mention the CDC guidelines for SSI. Including clinical and lab signs

2. Please highlight adjustment of the logistic regression for comorbidities

Reviewer #2: The manuscript titled: “Effect of oclacitinib (Apoquel) administration on development of surgical site infection following clean orthopedic stifle surgery” explores risk factors associated with specific types of surgeries in dogs. The main focus is on the effect of oclacitinib even though the effects of other variables are statistically more significant. As the authors point out the main drawback of this study is the retrospective design. I have some concerns about some results being presented with very little detail and a few other issues I think the authors need to address prior to publication.

Title: I would urge the authors to rethink the title. While very clear and concise, it does give the image of a clinical trial where one group has received oclacitinib and the other has not. The retrospective nature of the study should be mentioned.

Abstract: Since there was a very significant statistical correlation between anesthesia time and wound infection there would be much more sense in mentioning that first in the abstract.

Line 53-54: The fact that this minimizes the need for antibiotics could also be mentioned. Further, I am not sure about the “overall hospital cleanliness”, perhaps this could be rephrased to have something to do with hospital hygiene?

Line 103-104: Could an owner have taken their dog to another veterinary clinic due to SSI signs and thus have been left out? Is there a possibility of bias, e.g. an owner with a dog that develops SSI not trusting the veterinary hospital where the procedure was first performed and thus seeking help elsewhere? This could be discussed.

Line 111-112: How well was this prophylactic time adhered to? In my experience there is huge variation in when the antibiotic is administered (anywhere from a couple of hours before incision to a few minutes after).

Line 114-115: Why is a triple-antibiotic ointment used? If all goes to plan it shouldn’t be, correct?

Line 130-131: These are the factors selected that may have affected SSI, there are plenty more, such as body temperature during the procedure, other medications (e.g. corticosteroids), time of prophylactic antibiotics administration etc. Please rephrase.

Results: The results could be presented more clearly. E.g. the white blood cell count is not presented, only the p-values. This makes it impossible for the reader to interpret the results. I would ask the authors add these values (similar to the surgery time) for all variables investigated. Also, it could be preferrable to have the results presented in a table.

Line 177-179: Why are no p-values displayed?

Line 183-184: Why is the p-value not displayed here? Please add for consistency.

Discussion: While I agree with what the authors are saying here, I would still urge them to also look at this from a purely proportional perspective; an overall SSI rate in clean orthopedic procedures of 8/100 patients may still be manageable, but 19/100 is catastrophic and cause for serious concern. The authors correctly point out that the results may have been statistically significant if more cases with the treatment were discovered, but I think some discussion could be added from a more practical point-of-view rather than a statistical one.

6. PLOS authors have the option to publish the peer review history of their article (what does this mean?). If published, this will include your full peer review and any attached files.

Reviewer #1: **Yes: **Mahmoud Elfiky

Reviewer #2: **Yes: **Thomas Sven Christer Grönthal

---

## [Author Response · Author response to Decision Letter 0]

13 Jul 2023

PONE-D-23-10920

Effect of oclacitinib (Apoquel) administration on development of surgical site infection following clean orthopedic stifle surgery

Thank you for your review of my submitted article. I have reviewed and considered the comments provided and have addressed those comments below:

1. The supplied links to the PLOS ONE style templates were reviewed to assure the submitted manuscript meets the style requirements. I have reviewed the manuscript style and applied the files names as directed in the revision email. I did not identify any other style format issues. If there are additional style format errors that I have not identified, please let me know so I can resolve the issues.

2. There are no restrictions to providing the data set for this study. The data file will be uploaded as a supporting information file, along with the revised manuscript. 

3. I have reviewed my reference list. The list is complete and correct. On further review using PubMed, I have not identified any cited papers that have been retracted. If there is a certain reference that is of concern that I have not identified, please list the reference so I can further investigate if it should remain a relevant addition to my manuscript. 

Response to Reviewer Comments to the Author:

Reviewer #1: Thank you for your comments regarding my manuscript. I have addressed your comments below. Original comments have been listed, followed by a rebuttal statement.

1. Please mention the CDC guidelines for SSI. Including clinical and lab signs.

- I discussed the specifics of the CDC guidelines as used to identify patients in this study with surgical site infections. Lines 92-105 detail the CDC definitions of superficial, deep, and organ/space surgical site infections as defined by the CDC guidelines. As such, no changes were made to the manuscript addressing this comment.

2. Please highlight adjustment of the logistic regression for comorbidities

- Dogs with previously identified comorbidities were excluded from statistical analysis. The portion of the manuscript addressing reasons for patient exclusion has been updated to more clearly acknowledge this. See lines 81-83 of the revised manuscript.

Reviewer #2: Thank you for your comments regarding my manuscript. I have addressed your comments below. Original comments have been listed, followed by a rebuttal statement.

Title: I would urge the authors to rethink the title. While very clear and concise, it does give the image of a clinical trial where one group has received oclacitinib and the other has not. The retrospective nature of the study should be mentioned.

- The title of the manuscript has been changed to more clearly reflect the retrospective nature of this study as was requested. See lines 1-2.

Abstract: Since there was a very significant statistical correlation between anesthesia time and wound infection there would be much more sense in mentioning that first in the abstract.

- As suggested, the order of variables mentioned in the abstract was rearranged to mention the significance of anesthesia time first. See lines 37-38.

Line 53-54: The fact that this minimizes the need for antibiotics could also be mentioned. Further, I am not sure about the “overall hospital cleanliness”, perhaps this could be rephrased to have something to do with hospital hygiene?

- I have added a sentence explaining the importance of minimizing the use of antibiotics as it relates to the development of multidrug resistant infections. See lines 54-56. I have also deleted the phrase about hospital cleanliness. 

Line 103-104: Could an owner have taken their dog to another veterinary clinic due to SSI signs and thus have been left out? Is there a possibility of bias, e.g. an owner with a dog that develops SSI not trusting the veterinary hospital where the procedure was first performed and thus seeking help elsewhere? This could be discussed.

- Qualification for inclusion in this study required complete medical records including evaluation by the surgery service at both 2 and 8 weeks post operatively to reasonably avoid missing these types of cases. However, Lines 236-238 also does address your concern that infections may have been missed if they were seen by their primary veterinarian, treated, and record of that treatment was not forwarded to the surgery service for inclusion in our medical records. Additionally, because all patients included were seen for the requested 2 week and 8 week follow up appointments, this suggests that none of the included patients had owners who lost trust in hospital and sought definitive treatment elsewhere.

Line 111-112: How well was this prophylactic time adhered to? In my experience there is huge variation in when the antibiotic is administered (anywhere from a couple of hours before incision to a few minutes after).

- A check list is performed prior to initial incision of every surgery performed at our institution that includes confirmation that a dose of peri-operative antibiotics has been administered within the previous 60 minutes. This checklist is included as part of the medical record and antibiotic administration as described was confirmed for inclusion in this study. This has been more clearly stated in the manuscript, see lines 113-115.

Line 114-115: Why is a triple-antibiotic ointment used? If all goes to plan it shouldn’t be, correct?

- This is standard procedure at our hospital and applied once on a telfa pad following surgical procedures. I have adjusted the manuscript to clarify the ointment is applied once immediately post-op. See lines 117-119.

Line 130-131: These are the factors selected that may have affected SSI, there are plenty more, such as body temperature during the procedure, other medications (e.g. corticosteroids), time of prophylactic antibiotics administration etc. Please rephrase.

- Lines 51-54 acknowledge the multifactorial nature of surgical site infections. The manuscript was changed to denote the specific variables evaluated in this study. See lines 133-136. 

Results: The results could be presented more clearly. E.g. the white blood cell count is not presented, only the p-values. This makes it impossible for the reader to interpret the results. I would ask the authors to add these values (similar to the surgery time) for all variables investigated. Also, it could be preferable to have the results presented in a table.

- Range and median values were added for white blood cell count and neutrophil count to improve interpretation of the results. See lines 173-175 and lines 178-180.

- All data for this study has been provided with this revision. I feel the large number of patients in the study (n=390) prohibits presenting all variable values as a table within the manuscript.

Line 177-179: Why are no p-values displayed?

- P values were not originally included for outcomes that were statistically not significant, however have been added into this manuscript as requested. See lines 183-186.

Line 183-184: Why is the p-value not displayed here? Please add for consistency.

- P values were not originally included for outcomes that were statistically not significant, however have been added into this manuscript as requested. See lines 286-191.

Discussion: While I agree with what the authors are saying here, I would still urge them to also look at this from a purely proportional perspective; an overall SSI rate in clean orthopedic procedures of 8/100 patients may still be manageable, but 19/100 is catastrophic and cause for serious concern. The authors correctly point out that the results may have been statistically significant if more cases with the treatment were discovered, but I think some discussion could be added from a more practical point-of-view rather than a statistical one.

- The overall SSI rate in this study population was 8.7%, not 19%. The authors agree than an overall SSI rate of 19% could be considered catastrophic, it is still lower than some previously reported ranges in published literature as referenced in the manuscript. Additionally, in this study, the 19% SSI rate applies only to the group of dogs being treated with apoquel, and while we agree the rate is high and not ideal in any sense, statistically it did not prove significantly different from the no-apoquel group that had an SSI rate of 8.0%. I have added additional information in the study limitations section highlighting the large size difference between the group populations and how that may have affected the statistical analysis. See lines 239-241.

Thank you once again for your thoughtful input on this manuscript and hopefully I have addressed these concerns to your satisfaction.

---

## [Decision Letter · Decision Letter 1]

26 Jul 2023

A retrospective evaluation of the effect of oclacitinib (Apoquel) administration on development of surgical site infection following clean orthopedic stifle surgery

PONE-D-23-10920R1

Dear Dr. Motz,

We’re pleased to inform you that your manuscript has been judged scientifically suitable for publication and will be formally accepted for publication once it meets all outstanding technical requirements.

Kind regards,

Academic Editor

PLOS ONE

Additional Editor Comments (optional):

Reviewers' comments:

Reviewer's Responses to Questions

**Comments to the Author**

1. If the authors have adequately addressed your comments raised in a previous round of review and you feel that this manuscript is now acceptable for publication, you may indicate that here to bypass the “Comments to the Author” section, enter your conflict of interest statement in the “Confidential to Editor” section, and submit your "Accept" recommendation.

Reviewer #1: All comments have been addressed

Reviewer #2: All comments have been addressed

2. Is the manuscript technically sound, and do the data support the conclusions?

Reviewer #1: Yes

Reviewer #2: Yes

3. Has the statistical analysis been performed appropriately and rigorously? 

Reviewer #1: Yes

Reviewer #2: Yes

4. Have the authors made all data underlying the findings in their manuscript fully available?

Reviewer #1: Yes

Reviewer #2: Yes

5. Is the manuscript presented in an intelligible fashion and written in standard English?

Reviewer #1: Yes

Reviewer #2: Yes

6. Review Comments to the Author

Reviewer #1: I have seen the Author replying to all reviewers' comments.

I have one comment, coming from a Medical background; the CDC SSI guidelines are based on Human pathology. Applying them to Veterinary diseases requires a reference.

Reviewer #2: The authors have addressed all my comments, but I detected one very small detail; cell counts in the results should be presented as per mm^3. E.g. "3.74-11.9 mm3" should be "3.74-11.9 /mm3".

7. PLOS authors have the option to publish the peer review history of their article (what does this mean?). If published, this will include your full peer review and any attached files.

Reviewer #1: **Yes: **Mahmoud Elfiky

Reviewer #2: **Yes: **Thomas Grönthal

---

## [Editor Report · Acceptance letter]

31 Jul 2023

PONE-D-23-10920R1 

A retrospective evaluation of the effect of oclacitinib (Apoquel) administration on development of surgical site infection following clean orthopedic stifle surgery 

Dear Dr. Motz:

I'm pleased to inform you that your manuscript has been deemed suitable for publication in PLOS ONE. Congratulations! Your manuscript is now with our production department. 

Kind regards, 

on behalf of

Dr. Robert Jeenchen Chen 

Academic Editor

PLOS ONE